# Bottom-up and top-down influences at untrained conditions determine perceptual learning specificity and transfer

Ying-Zi Xiong[1], Jun-Yun Zhang[1], Cong Yu[1,2,3]*

[1]School of Psychological and Cognitive Sciences, Peking University, Beijing, China; [2]IDG/McGovern Institute for Brain Research, Peking University, Beijing, China; [3]Peking-Tsinghua Center for Life Sciences, Peking University, Beijing, China

**Abstract** Perceptual learning is often orientation and location specific, which may indicate neuronal plasticity in early visual areas. However, learning specificity diminishes with additional exposure of the transfer orientation or location via irrelevant tasks, suggesting that the specificity is related to untrained conditions, likely because neurons representing untrained conditions are neither bottom-up stimulated nor top-down attended during training. To demonstrate these top-down and bottom-up contributions, we applied a "continuous flash suppression" technique to suppress the exposure stimulus into sub-consciousness, and with additional manipulations to achieve pure bottom-up stimulation or top-down attention with the transfer condition. We found that either bottom-up or top-down influences enabled significant transfer of orientation and Vernier discrimination learning. These results suggest that learning specificity may result from under-activations of untrained visual neurons due to insufficient bottom-up stimulation and/or top-down attention during training. High-level perceptual learning thus may not functionally connect to these neurons for learning transfer.

*For correspondence: yucong@ pku.edu.cn

**Competing interests:** The authors declare that no competing interests exist.

## Introduction

Visual perceptual learning is the process in which the observers improve their discrimination of fine differences of basic visual features, such as contrast, orientation, motion direction, etc., through practice. For several decades visual perceptual learning has been regarded as a distinct format of learning because it is specific to the orientation and retinal location of the trained stimulus (*Schoups et al., 1995*; *Ahissar and Hochstein, 1997*; *Shiu and Pashler, 1992*; *Dosher and Lu, 1998*; *Poggio et al., 1992*; *Fiorentini and Berardi, 1980*; *Yu et al., 2004*). Such learning specificity has inspired theories that interpret visual perceptual learning as a result of training induced neural plasticity in the early visual areas (*Schoups et al., 1995*; *Karni and Sagi, 1991*; *Teich and Qian, 2003*; *Bejjanki et al., 2011*). For example, it has been proposed that training could sharpen neuronal orientation tuning in the primary visual cortex (V1), so that neurons become more sensitive to the fine changes of orientation differences (*Teich and Qian, 2003*; *Schoups et al., 2001*). The learning specificity has also constrained alternative reweighting theories of visual perceptual learning (*Dosher and Lu, 1998*; *Poggio et al., 1992*; *Yu et al., 2004*; *Mollon and Danilova, 1996*; *Petrov et al., 2005*; *Law and Gold, 2008*; *2009*). These theories propose that training may not change the tuning properties of sensory neurons. Rather, the inputs from activated neurons are reweighted through training to improve readout at a later decision stage. Reweighting theories are supported by neurophysiological evidence. For example, motion direction learning in monkeys is

**eLife digest** People can become more sensitive to small changes in what they are seeing – such as detecting a slight change in the angle of a particular line – with practice. This process is called perceptual learning, but the improvement is often specific such that it is typically lost if the line moves to a new place, or a different line angle is used. Previous work does show that it is possible to transfer the learning to a new location or angle if the individual also practices another, seemingly irrelevant, task at the same or a later time – such as judging how bright the line is.

To understand what might be happening to produce these seemingly conflicting results, Xiong et al. used a technique called "continuous flash suppression" with human volunteers. This approach meant that the volunteers were shown an object (such as an angled line) in one eye, while their other eye viewed white noise similar to the "snowflakes" seen on an old-fashioned un-tuned television screen. The flashing snowflakes in one eye meant that the volunteers were not consciously aware of the presence of the angled line in the other eye.

The experiments revealed that perceptual learning at the new location or line angle happened when a subconsciously-observed object was shown in the new location or angle, or when the volunteers were asked to pay attention to the "subconscious object" when no object was actually there. This suggests that perceptual learning can happen in new conditions both through 'bottom-up' processes, which rely entirely on information coming in from the senses, and 'top-down' processes, which are influenced by what a person is aware of and paying attention to. What is more, the results suggest that the classical observations of specificity in perceptual learning are likely to be a result of the lack of bottom-up and top-down influences in the untrained condition, when the volunteers work hard to improve their performance with the trained condition.

Future studies could directly look at what is going on in the brain when perceptual learning becomes less specific, for example by using a technique like functional magnetic resonance imaging to measure brain activity.

correlated with changes in motion-driven responses of neurons in the lateral intraparietal area (LIP) that is related to the transformation of motion information into decisions (saccadic choices), but not with changes of neurons in the medial temporal area (MT) that represents motion direction signals (*Law and Gold, 2008*).

However, in a series of 'double training' studies we demonstrated that visual perceptual learning of various tasks can transfer significantly, and often completely, to new orientations or locations (*Xiao et al., 2008*; *Wang et al., 2012*, *2014*; *Zhang et al., 2010*, *2014*). In a double training experimental design the observers are additionally exposed to the new orientation or location via practicing an irrelevant task besides the primary learning task. For example, perceptual learning of foveal orientation discrimination (e.g., which of two consecutively presented gratings is more clockwise-tilted?) initially shows little transfer to an orthogonal orientation. However, if the observers are exposed to an orthogonal orientation via an irrelevant contrast discrimination task (e.g., which of two gratings has higher contrast?), learning transfers completely to the orthogonal orientation (*Zhang et al., 2010*). Similarly, perceptual learning of Vernier discrimination (e.g., whether a lower grating is placed to the left or right of an upper grating) can also transfer significantly and often completely to a new retinal location when the observers are additionally exposed to the transfer location via an irrelevant contrast or orientation discrimination task (*Wang et al., 2012*, *2014*). These results suggest two important insights regarding visual perceptual learning. First, visual perceptual learning is mainly a high-level rule-based learning process that occurs beyond the retinotopic and orientation selective visual areas, so that learning is in principle transferrable to untrained conditions (*Zhang et al., 2010*). Second, learning specificity may be related to the untrained conditions, rather than the trained conditions as the field has been assuming (i.e., plasticity with the trained early visual cortical neurons or reweighting of the inputs from these neurons). The second insight, which stands completely different from the interpretations of specificity by the field, forms the basis of the current study.

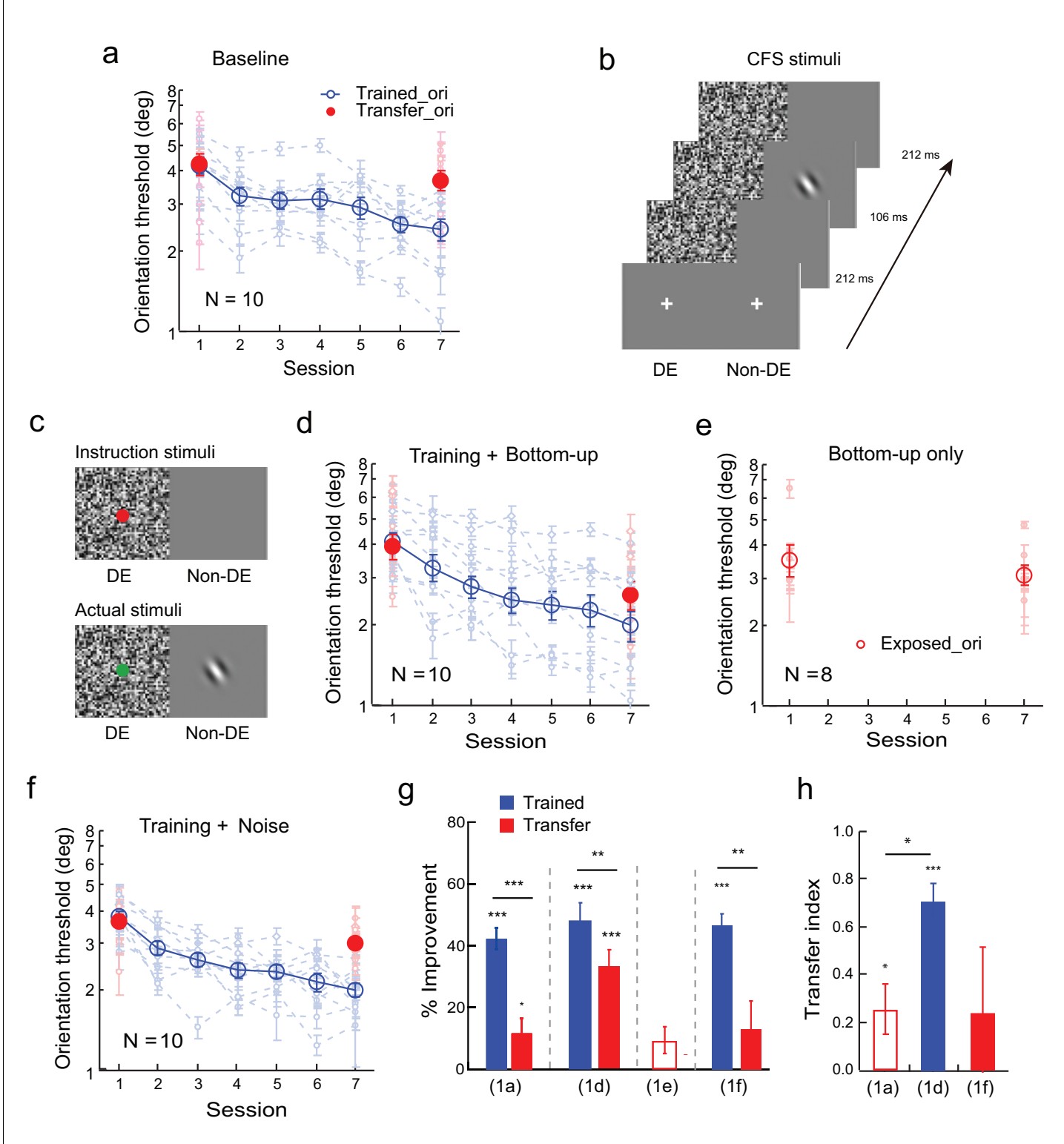

**Figure 1.** Orientation discrimination learning and the effect of bottom-up stimulation of the transfer orientation on learning transfer. (**a**) The baseline. Training was conducted at one orientation and the transfer of learning was tested at an orthogonal orientation to reveal orientation specificity. (**b**) CFS configurations for subconscious presentation of a Gabor at the orthogonal transfer orientation. The flashing noise was presented from 0 to 530 ms in the dominant eye. From 212 to 318 ms, a Gabor at the transfer orientation was also presented in the non-dominant eye. (**c**) Stimuli for bottom-up stimulation. Bottom-up stimulation was achieved by different CFS stimuli for instruction presentations before the experiment and for actual experimental presentations. Instruction stimuli: Before the experiment, the observers were instructed to report the color of a dot centered on the noise

*Figure 1 continued on next page*

*Figure 1 continued*

pattern in the dominant eye. A blank screen was shown to the non-dominant eye. Actual stimuli: In the actual experiment, a Gabor at the orthogonal transfer orientation was also flashed in the non-dominant eye. The observers were not told, neither were they aware of, the presence of the orthogonal Gabor. Here orientation discrimination training and bottom-up stimulation of the orthogonal transfer orientation were performed in different blocks of trials. (d) The mean and individual learning and transfer data with training and bottom-up stimulation of the transfer orientation. (e) Control experiment. Same as 1d except that there was no orientation discrimination training. (f) Control experiment. Same as 1d except that there was no actual presence of the orthogonal Gabor. (g) A summary of learning and transfer in the baseline, training plus bottom-up stimulation, bottom-up stimulation alone, and training plus noise-only conditions. (h) A summary of the transfer indices in the baseline, training plus bottom-up stimulation, and training plus noise-only conditions. Error bars indicate ± 1 standard error of the mean. DE - dominant eye. *p<0.05; **p<0.01; ***p<0.001. See *Figure 1—source data 1* for raw data.

The following source data is available for figure 1:

**Source data 1.** The first data sheet summarizes the mean and individual data presented in figure panels 1a, 1d, 1e, and 1f.

During training at a specific orientation or location in a typical visual perceptual learning experiment, most mental resources are devoted to the trained stimuli. For example, an observer has to focus the attention on the near-threshold difference between the reference orientation and the target orientation at a specific location. As a result the untrained orientations and locations, and thus the visual neurons representing these orientations and locations, are neither bottom-up stimulated nor top-down attended during training. We thus suspect that this insufficient bottom-up stimulation of, and/or top-down attention to, the untrained conditions are responsible for the orientation and location specificity. In other words, because visual neurons representing untrained conditions are not properly activated as a result, high-level perceptual learning may not be able to functionally connect to these neurons for learning transfer.

Our previous double training experiments are unable to separately identify the potential bottom-up and/or top-down contributions because suprathreshold stimuli are used in the secondary exposure task. In the current study we applied a continuous flash suppression (CFS) technique (*Tsuchiya and Koch, 2005*) to suppress the exposure stimulus into sub-consciousness (see Materials and methods). We further manipulated the subconscious stimulus conditions to make the exposure task to be bottom-up only or top-down only. The results show that either bottom-up stimulation of the untrained condition, or top-down attention to it, is sufficient to enable substantial and often complete transfer of learning. These results provide a solution to the mystery of learning specificity that has dominated the history of perceptual learning research. With learning specificity considered as a by-product of training, the field should move on to study the brain mechanisms of perceptual learning without much of specificity-related constraints. Moreover, more efficient training paradigms can be designed to generate perceptual learning without the unwanted specificity in practical settings.

## Results

### Orientation specificity and transfer: The effects of bottom-up or top-down influences at the untrained orientations

#### Baseline: Orientation specificity

We first established the baseline for orientation specificity in an orientation discrimination learning task. The observers practiced orientation discrimination with a foveal Gabor stimulus at 36° or 126° for 5 daily sessions, which reduced the thresholds by $42.1 \pm 3.4\%$ (mean ± se; $t_9 = 12.46$, p<0.001, 95% CI = 34.4% to 49.7%, Cohen's d = 3.94; two-tailed paired t-test in this and later analyses when the significance of the improvement was tested) at the trained orientation (*Figure 1a and g*). Training only reduced the orientation thresholds with the same stimulus at the untrained orthogonal orientation by $11.8 \pm 4.6\%$ ($t_9 = 2.56$, p = 0.030, 95% CI = 1.4% to 22.2%, Cohen's d = 0.81). The improvement was significantly lower at the untrained orientation than at the trained orientation ($t_9 = 6.39$, p<0.001, 95% CI = 19.56% to 40.98%, Cohen's d = 2.02). We used a transfer index (TI) to compare the transfer effects among various conditions. The transfer index was the improvement at the transfer condition divided by the improvement at the trained condition, with TI = 0 indicating

complete learning specificity, and TI = 1 indicating complete learning transfer. Here TI = 0.25 ± 0.10 ($t_9$ = 2.47, p = 0.036, 95% CI = 0.02 to 0.49, Cohen's d = 0.78; *Figure 1h*), indicating that learning is mostly orientation specific.

## The effects of bottom-up exposure of the untrained orientation on learning transfer

Previously we have shown that orientation discrimination learning can completely transfer to an orthogonal orientation if the observers receive additional exposure to the transfer orientation via an irrelevant secondary task such as contrast discrimination (*Zhang et al., 2010*). To separate the potential bottom-up and top-down contributions of the secondary task, we applied a continuous flashing suppression (CFS) technique (*Tsuchiya and Koch, 2005*) to render the exposure stimulus (a Gabor orthogonal to the trained orientation) subconscious. This was achieved by presenting the flashing noise to the dominant eye, which suppressed the perception of the orthogonal Gabor in the non-dominant eye (*Figure 1b*).

We further manipulated the CFS stimulus configuration to achieve bottom-up stimulation of the orthogonal transfer orientation. Before the experiment, we instructed ten new observers to report the color (red/green) of a dot that was centered on the noise pattern (*Figure 1c* instruction stimuli; the mean correct rate was 96.6 ± 0.3%) in the dominant eye while showing a blank screen to the non-dominant eye. But in the actual experiment an orthogonal Gabor was present in the non-dominant eye (*Figure 1c* actual stimuli). Since the observers were neither told, nor aware of, the existence of the orthogonal Gabor, and their attention was directed to the dot-color report task, the subconsciously presented orthogonal Gabor bottom-up stimulated the visual cortical neurons at the transfer orientation without attracting top-down attention.

The observers received orientation training and bottom-up stimulation of the orthogonal transfer orientation in separate blocks of trials in the same session. Five sessions of practice reduced the orientation discrimination thresholds at the trained orientation significantly by 50.6 ± 6.0% ($t_9$ = 8.87, p<0.001, 95% CI = 35.9% to 60.5%, Cohen's d = 2.80), as well as at the untrained but bottom-up stimulated orthogonal orientation by 33.7 ± 4.6% ($t_9$ = 7.33, p<0.001, 95% CI = 23.3% to 44.1%, Cohen's d = 2.32) (*Figure 1d and g*). There was still a significant difference between the two improvements ($t_9$ = 3.84, p = 0.004, 95% CI = 5.9% to 23.0%, Cohen's d = 1.21), and the transfer index TI = 0.72 ± 0.08 ($t_9$ = 9.09, p<0.001, 95% CI = 0.54 to 0.90, Cohen's d = 2.87; *Figure 1h*), indicating substantial but incomplete learning transfer.

A control experiment indicated that the improvement at the untrained orthogonal orientation did not result from the bottom-up stimulation alone. Eight new observers received equal amount of bottom-up stimulation of the orthogonal transfer orientation without practicing the orientation discrimination task. The bottom-up stimulation changed the orientation discrimination thresholds at the stimulated orientation insignificantly by 9.7 ± 4.3% ($t_7$ = 2.25, p = 0.06, 95% CI = -0.05% to 19.9%, Cohen's d = 0.85, *Figure 1e and g*).

A second control experiment ruled out the possibility that the dynamic white noise, which would activate visual neurons tuned to all orientations, was sufficient to enable the same amount of learning transfer. The experimental design was the same as in *Figure 1c*, except that no orthogonal Gabor was present in the actual stimuli. Again training reduced orientation thresholds significantly by 46.7 ± 3.4% ($t_9$ = 13.92, p<0.001, 95% CI = 39.1% to 54.3%, Cohen's d = 4.40) in ten observers (*Figure 1f and g*). However, no significant threshold improvement was evident at the orthogonal transfer orientation (13.4 ± 8.8%, $t_9$ = 1.53, p = 0.16, 95% CI = -6.5% to 33.2%, Cohen's d = 0.51), which was significantly lower than the improvement at the trained orientation ($t_9$ = 4.13, p = 0.003, 95% CI = 15.1% to 51.6%, Cohen's d = 1.44). The TI = 0.24 ± 0.24 in this condition ($t_9$ = 0.99, p = 0.35, 95% CI = -0.31 to 0.79, Cohen's d = 0.31; *Figure 1h*). The error bar of the transfer index was large because one observer actually showed complete learning transfer.

An independent-samples Kruskal-Wallis test revealed significant differences among the transfer indices of the baseline condition (*Figure 1a*), the training plus bottom-up stimulation condition (*Figure 1d*), and the training plus noise-only condition (*Figure 1f*) (p = 0.009). Post-hoc Dunn's multiple comparison indicated that the training plus bottom-up stimulation condition had significantly more transfer than the baseline condition (p = 0.004 without correction; p = 0.011 with correction) and the training plus noise-only condition (p = 0.022 without correction or 0.067 with correction).

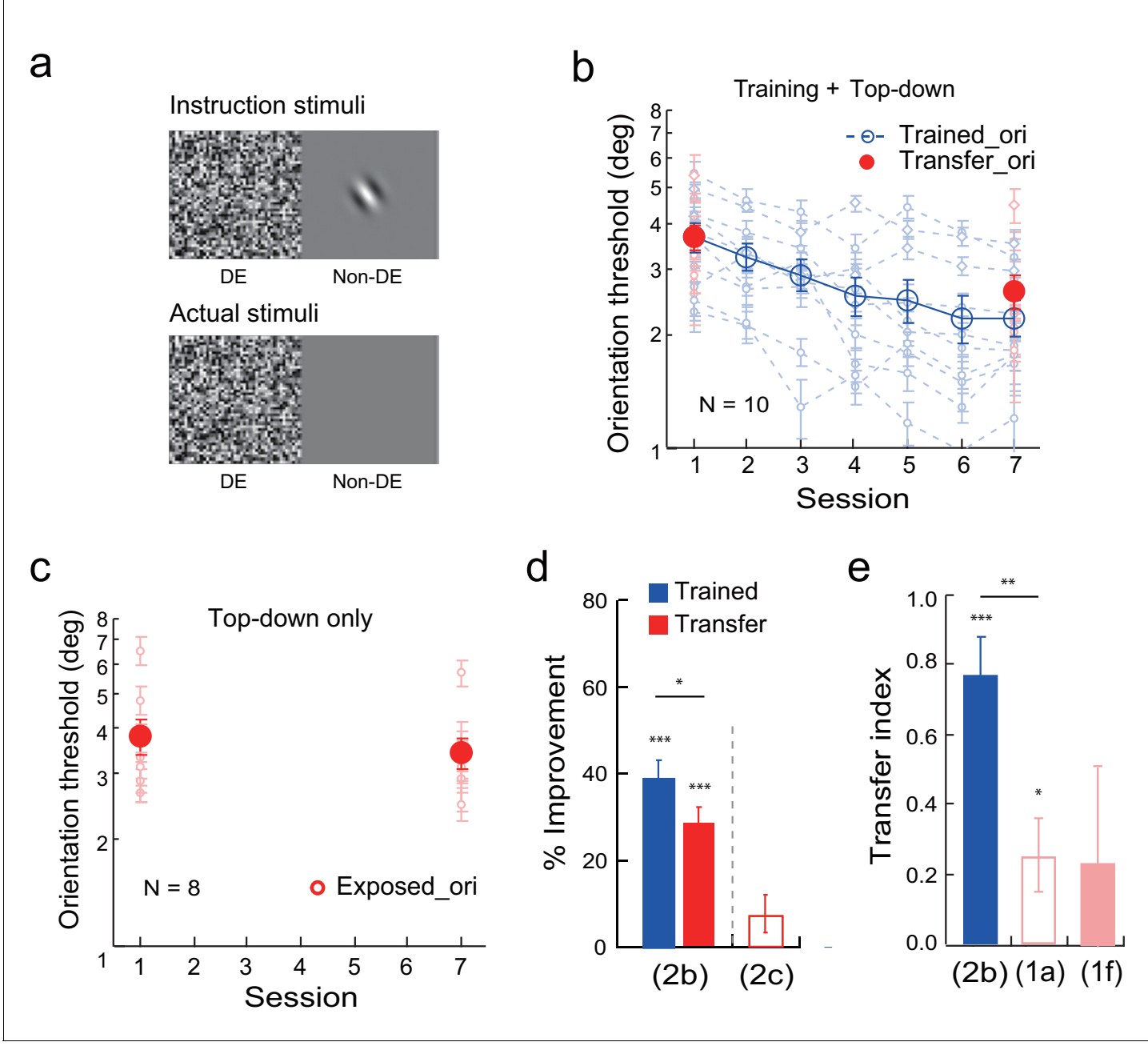

**Figure 2.** Orientation discrimination learning and the effect of top-down attention to the transfer orientation on learning transfer. (a) CFS configurations for the top-down attention condition. Instruction stimuli: Before the experiment the observers were instructed through a demo that an orthogonal Gabor or an uppercase letter C would be presented to the non-dominant eye, and that they needed to report whether it was a Gabor or a C even if they could not perceive it. Actual stimuli: In actual testing the Gabor/C was replaced by a blank screen. Orientation discrimination training and the top-down attention condition were conducted in different blocks of trials. (b) The mean and individual learning and transfer data with training and top-down attention to the transfer orientation. (c) Control experiment. Same as 2b except that there was no orientation discrimination training. (d) A summary of learning and transfer in the top-down attention and control conditions. (e) A summary of the transfer indices in the current training plus top-down attention condition and the previous baseline (replotted from *Figure 1a*) and training plus noise-only (replotted from *Figure 1f*) conditions. Error bars indicate ± 1 standard error of the mean. DE - dominant eye. *p<0.05; **p<0.01; ***p<0.001. See *Figure 2—source data 1* for raw data.

The following source data is available for figure 2:

**Source data 1.** The first data sheet summarizes the mean and individual data presented in figure panels 2b and 2c.

There was no significant difference of the transfer effects between the training plus noise-only condition and the baseline condition (p = 0.54 without correction or 1.00 with correction).

There results together suggest that bottom-up stimulation of the transfer orientation can enable substantial but partial transfer of orientation learning. Moreover, high-level perceptual learning may functionally connect to salient orientation signals for learning transfer even if these signals are subconscious.

## The effects of top-down attention to the untrained orientation on learning transfer

Next we studied the effect of top-down attention to the transfer orientation on learning transfer. To isolate the top-down effect, before the experiment the observers were shown the flashing noise in the dominant eye. They were also shown a Gabor at the orthogonal transfer orientation, or an uppercase letter C, in the non-dominant eye when the other eye was closed (*Figure 2a* instruction stimuli). They were asked to report, or guess if they had to, whether a Gabor or a letter C was shown every trial by key press. However, during actual training experiment neither the Gabor nor the letter C was presented (*Figure 2a* actual stimuli). Faked feedback was provided to give the observers a false impression that 70–80% of their responses were correct in each block of trials. This stimulus manipulation thus initiated top-down attention to the orthogonal transfer orientation without actual bottom-up stimulation.

Training and top-down attention to the transfer orientation improved orientation thresholds at the trained orientation by 39.5 ± 3.5% ($t_9$ = 11.21, p<0.001, 95% CI = 31.5% to 47.4%, Cohen's d = 3.54) in ten observers (*Figure 2b and d*). The threshold reduction at the orthogonal orientation was also significant (29.0 ± 3.4%, $t_9$ = 8.45, p<0.001, 95% CI = 21.3% to 36.8%, Cohen's d = 2.67), indicating the effectiveness of top-down orientation attention in enabling learning transfer. The improvement at the untrained orientation was significantly lower than that at the trained orientation ($t_9$ = 2.53, p = 0.032, 95% CI = 1.1% to 19.8%, Cohen's d = 0.80), and the transfer index was 0.77 ± 0.10 ($t_9$ = 7.86, p<0.001, 95% CI = 0.55 to 1, Cohen's d = 2.48; *Figure 2e*), suggesting substantial but partial learning transfer.

A control experiment with eight observers indicated that, without actual training of orientation discrimination, top-down attention to the transfer orientation alone led to an insignificant reduction of orientation threshold at the transfer orientation (8.0 ± 4.4%, $t_7$ = 1.81, p = 0.11, 95% CI = -2.4% to 18.4%, Cohen's d = 0.64) (*Figure 2c and d*). This control ruled out the possibility that top-down attention alone improved orientation discrimination at the transfer orientation.

An independent-samples Kruskal-Wallis test revealed significant differences among the transfer indices of the baseline condition (*Figure 1a*), the training plus noise-only condition (*Figure 1f*), and the training plus top-down attention condition (p = 0.007). Post-hoc Dunn's multiple comparison indicated that the training plus top-down attention condition had significantly more transfer than the baseline condition (p = 0.003 without correction; p = 0.008 with correction) and the training plus noise-only condition (p = 0.019 without correction or 0.058 with correction). These results indicate that top-down attention, like bottom-up stimulation, is sufficient to enable substantial but partial learning transfer to the untrained orthogonal orientation.

## The effects of combined bottom-up stimulation and top-down attention to the untrained orientation on learning transfer

We eventually achieved complete learning transfer with combined bottom-up stimulation and top-down attention with the transfer orientation. The stimulus configuration was identical to the above top-down attention configuration, except that the orthogonal Gabor at the transfer orientation was indeed present in the actual experiment for 80% of the trials (*Figure 3a*), and the upper case C was present for the remaining 20%. The observers needed to guess whether a Gabor or a C was present, and auditory feedback was given on incorrect responses.

Training significantly reduced the orientation thresholds at both the trained orientation (45.1 ± 5.4%, $t_8$ = 8.25, p<0.001, 95% CI = 32.5% to 57.8%, Cohen's d = 2.49) and the orthogonal transfer orientation (39.0 ± 4.3%, $t_8$ = 9.00, p<0.001, 95% CI = 29.0% to 49.0%, Cohen's d = 2.38) in nine observers (*Figure 3b–d*). There was no significant difference between the two improvements ($t_8$ = 1.20, p = 0.26, 95% CI = -5.7% to 18.0%, Cohen's d = 0.40). The transfer index was 0.94 ± 0.12 ($t_8$ =

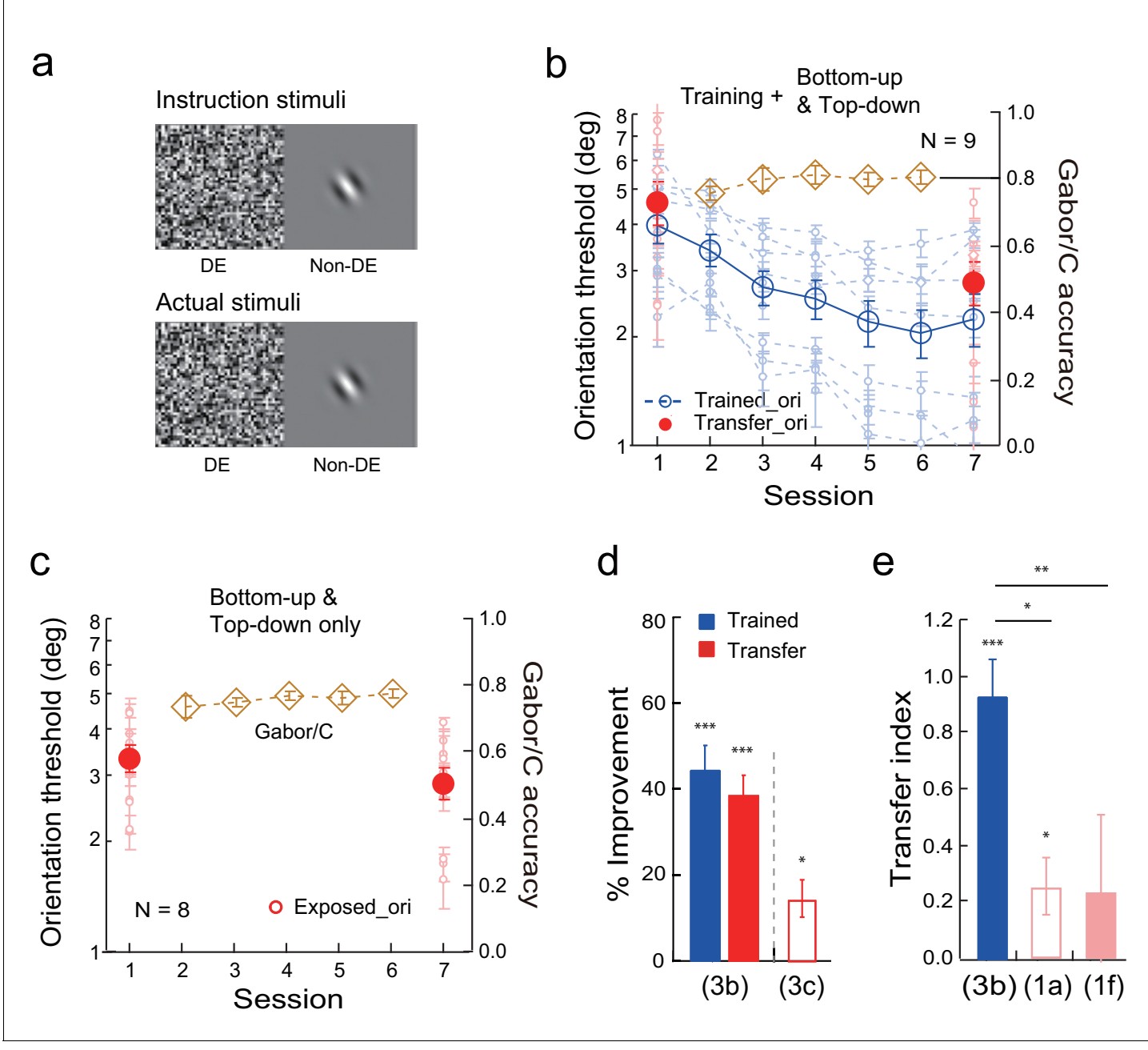

**Figure 3.** Orientation discrimination learning and the effect of combined bottom-up stimulation and top-down attention on learning transfer. (a) CFS configurations for the combined bottom-up stimulation and top-down attention condition. It only differed from *Figure 2a* in that the orthogonal Gabor was present in 80% of the trials in the actual stimuli, and an uppercase letter C was present in the remaining 20%. (b) The mean and individual learning and transfer data with training and combined bottom-up stimulation and top-down attention to the transfer orientation. (c) Control experiment. Same as 3b except that there was no orientation discrimination training. (d) A summary of learning and transfer in the combined bottom-up and top-down condition and the control condition. (e) A summary of the transfer indices in the current combined bottom-up and top-down condition and the previous baseline (replotted from *Figure 1a*) and training plus noise-only (replotted from *Figure 1f*) conditions. Error bars indicate ± 1 standard error of the mean. DE - dominant eye. *p<0.05; **p<0.01; ***p<0.001. See *Figure 3—source data 1* for raw data.

The following source data is available for figure 3:

**Source data 1.** The first data sheet summarizes the mean and individual data presented in figure panels 3b and 3c.

7.78, p<0.001, 95% CI = 0.66 to 1.22, Cohen's d = 2.59; *Figure 3e*), indicating nearly complete learning transfer to the untrained orthogonal orientation.

In a control experiment, the combined bottom-up stimulation and top-down attention without actual orientation training only led to a small threshold improvement at the exposed orientation (14.6 ± 4.4%, $t_7$ = 3.31, p=0.013, 95% CI = 4.2% to 25.1%, Cohen's d = 1.17) (*Figure 3c and d*). The improvement was significantly lower than that in *Figure 3b* with orientation training ($t_{15}$ = 3.93, p=0.001, 95% CI = 11.2% to 37.6%, Cohen's d = 1.91, two-tailed unpaired t-test). This control ruled out the possibility that the combined bottom-up stimulation and top-down attention to the transfer orientation alone caused the substantial improvement of orientation discrimination shown in *Figure 3b*. In *Figure 3b and c* the Gabor/C judgments were near chance (0.80) after some initial increase, suggesting effective suppression of stimulus perceptions with continuous flashing noise.

An independent-samples Kruskal-Wallis test revealed significant differences among the transfer indices of the baseline condition (*Figure 1a*), the training plus noise-only condition (*Figure 1f*), and the training plus combined bottom-up and top-down condition (p = 0.002). Post-hoc Dunn's multiple comparison indicated that the training plus combined bottom-up and top-down condition had significantly more transfer than the baseline condition (p = 0.001 without correction; p = 0.003 with correction) and the training plus noise-only condition (p = 0.007 without correction or 0.022 with correction).

In *Figure 3b and c* the Gabor/C judgments were near chance (0.80), suggesting effective suppression of the perception of the orthogonal Gabor by the continuous flashing noise. We further confirmed this suppression effect with 18 observers (10 from *Figure 1d*, 3 from *Figure 2b*, and 5 from *Figure 3b*) after they completed their experiments. Each observer was presented a sub-conscious Gabor at the transfer orientation in half the trials with flashing noise presented to the other eye (50 trials per block for four blocks). In the other half trials a blank screen was presented. The result showed a chance-level accuracy at 0.51 ± 0.01. Therefore, the transfer results in *Figure 1– 3* were not contaminated by the leakage of the Gabor perception.

## Location specificity and transfer: The effects of bottom-up and top-down influences at the untrained locations

Vernier learning is highly location specific. In our previous study (*Wang et al., 2014*) that used the same stimulus configuration as here in *Figure 4*, Vernier learning at one visual quadrant location showed zero transfer to a diagonal location, with the transfer index TI = -0.1 ± 0.16. However, additional training at the transfer location with an orientation discrimination task successfully enabled Vernier learning transfer (TI = 0.98 ± 0.22). The strong location specificity as well as the complete learning transfer after double training formed the baselines for the current experiments.

### The effects of bottom-up exposure of the untrained location on learning transfer

To investigate the effect of bottom-up stimulation of the transfer location on Vernier learning transfer, we asked the observers to perform the Vernier task while a Gabor was simultaneously presented at a diagonal quadrant location subconsciously. Specifically, the observers practiced the Vernier task in one visual quadrant of the dominant eye, while the opposite visual hemifield across the horizontal meridian was covered by flashing noise (*Figure 4a* instruction stimuli). In the actual experiment a horizontal Gabor was simultaneously presented at the diagonal location in the non-dominant eye (*Figure 4a* actual stimuli). Because of the near-threshold Vernier discrimination at the trained location and the noise suppression, this horizontal Gabor bottom-up stimulated the transfer location without the observers' awareness.

Eight observers received five sessions of Vernier training and simultaneous subconscious bottom-up stimulation of the diagonal transfer location. This procedure reduced Vernier thresholds by 32.4% ± 4.3% ($t_7$ = 7.54, p<0.001, 95% CI = 22.2% to 42.5%, Cohen's d = 2.32) at the trained location, as well as by 35.3% ± 6.4% ($t_7$ = 5.53, p<0.001, 95% CI = 20.2% to 50.4%, Cohen's d = 1.95) at the transfer location (*Figure 4b*). The two improvements were comparable ($t_7$ = 0.54, p = 0.60, 95% CI = −15.6% to 9.8%, Cohen's d = 0.19). The TI = 1.13 ± 0.17 ($t_7$ = 6.54, p<0.001, 95% CI = 0.72 to 1.55, Cohen's d = 2.31; *Figure 4f*), indicating complete transfer of Vernier learning. This learning transfer was not caused by the bottom-up stimulation alone. Without Vernier training (eight

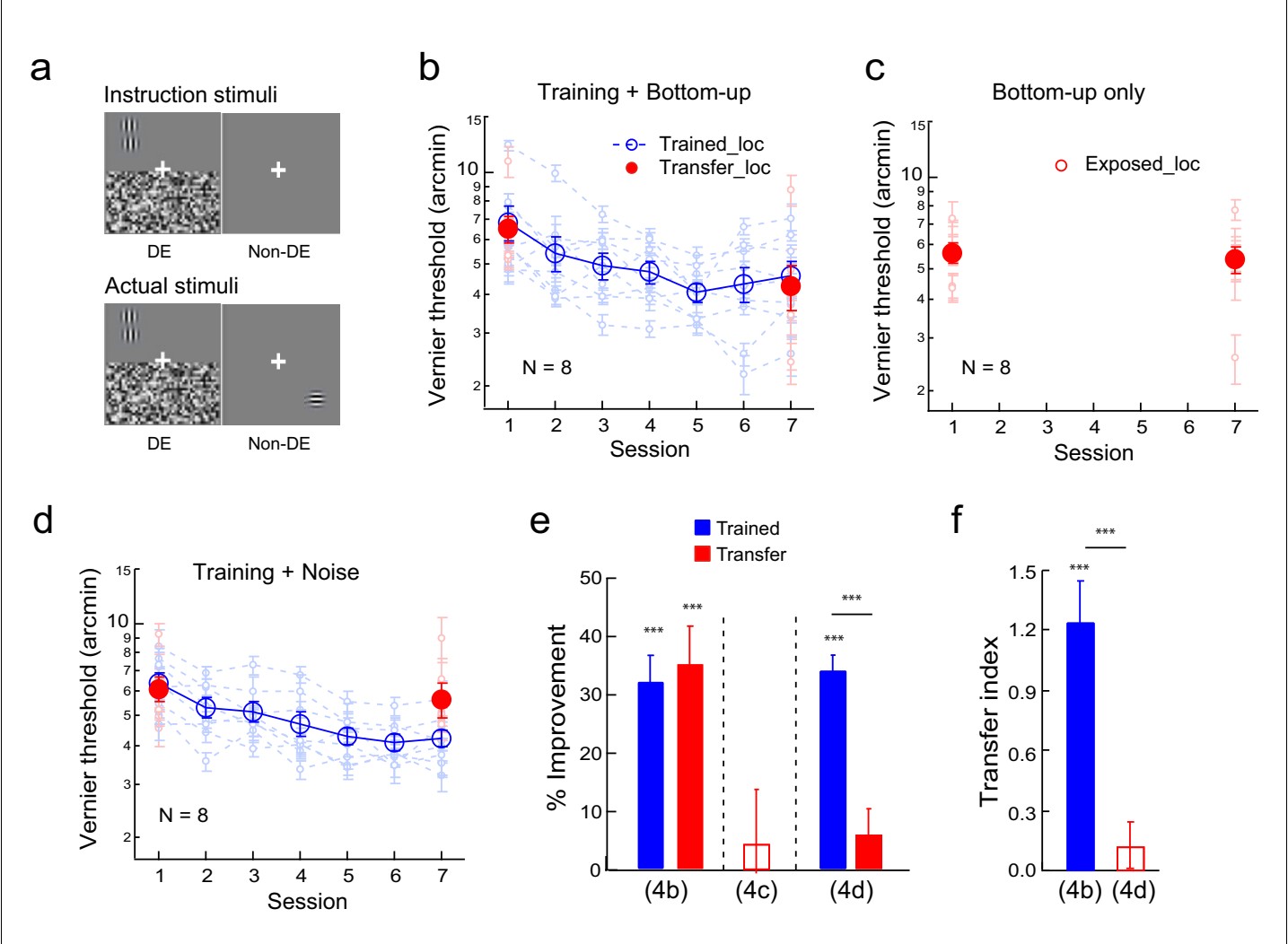

**Figure 4.** The effects of bottom-up stimulation of the transfer location on Vernier learning transfer. (a) CFS configurations for the simultaneous Vernier training and bottom-up stimulation of the diagonal transfer location. Instruction stimuli: The observer was instructed before the experiment to perform the Vernier task in the dominant eye, while flashing noise patterns covered the opposite visual hemifield across the horizontal meridian. A blank screen was shown to the non-dominant eye. Actual stimuli: A horizontal Gabor was simultaneously flashed in the non-dominant eye at the diagonal transfer location. The observers were neither told, nor were they aware of, the presence of the Gabor stimulus. (b) The mean and individual learning and transfer data with training and bottom-up stimulation of the transfer location. (c) Control experiment. Same as 4b except that there was no Vernier training. (d). Control experiment. Same as 4b except that there was no presence of the Gabor stimulus at the transfer location. (e). A summary of learning and transfer in the training plus bottom-up stimulation condition, the bottom-up stimulation alone condition, and the training plus noise-only condition. (f). A summary of the transfer indices in the training plus bottom-up stimulation condition and the training plus noise-only condition. Error bars indicate ± 1 standard error of the mean. DE - dominant eye. *p<0.05; **p<0.01; ***p<0.001. See *Figure 4—source data 1* for raw data.

The following source data is available for figure 4:

**Source data 1.** The first data sheet summarizes the mean and individual data presented in figure panels 4b, 4c, and 4d.

observers in a control condition instead reported the red/green color of a peripheral dot at the original Vernier stimulus location), mere bottom-up stimulation of the diagonal location produced insignificant change of Vernier performance at the stimulated location (4.1 ± 9.1%, $t_7$ = 0.45, p = 0.67, 95% CI = −17.5% to 25.7%, Cohen's d = 0.16) (*Figure 4c*).

We also tested whether the bottom-up location stimulation by the Gabor stimulus was necessary for the learning transfer to the untrained location, since a large noise field covering two visual

quadrants also included the untrained location. The experimental conditions were unchanged as in *Figure 4a* except that the horizontal Gabor was absent at the transfer location. The training reduced Vernier thresholds in eight new observers at the trained location by 32.3 ± 3.4% ($t_7$ = 9.42, p<0.001, 95% CI = 24.2% to 40.5%, Cohen's d = 3.33) (*Figure 4d*). However, training reduced Vernier thresholds at the transfer location insignificantly by 6.0 ± 4.3% ($t_7$ = 1.40, p = 0.21, 95% CI = −4.1% to 16.1%, Cohen's d = 0.49). The improvement was significantly lower at the transfer location than at the trained location ($t_7$ = 8.01, p<0.001, 95% CI = 18.6% to 34.2%, Cohen's d = 2.83). The transfer index TI = 0.13 ± 0.11 ($t_7$ = 1.16, p = 0.28, 95% CI = −13.5% to 39.7%, Cohen's d = 0.41; *Figure 4f*).

An independent samples Mann-Whitney U test revealed a significant difference of transfer indices between the training plus bottom-up stimulation condition (*Figure 4b*) and the training plus noise-only condition (p<0.001). The above results together indicated that bottom-up stimulation of the untrained location was indeed necessary as well as sufficient for the complete learning transfer shown in *Figure 4b*.

## The effects of top-down attention to the untrained location on learning transfer

To create a top-down stimulus configuration, the observers were instructed before the experiment to report/guess a Gabor or an uppercase C at the transfer location in the non-dominant eye while the noise was flashed in the dominant eye (*Figure 5a* instruction stimuli. The observers needed to close the dominant eye to see the Gabor or letter C). But the actual stimuli contained no Gabor or letter C (*Figure 5a* actual stimuli), which the observer was not aware of due to flashing noise suppression. Therefore during these trials, which alternated with Vernier discrimination trials in the same blocks, the observers top-down attended to the transfer location without actual bottom-up stimulation. Fake feedback was provided to give the observers an impression that their responses were 70–80% correct in a specific block. Prior to each block, a black square and a white square were shown to indicate the training and transfer locations, respectively.

The Vernier training and top-down attention to the transfer location improved Vernier thresholds at the trained location by 28.4 ± 4.0% ($t_7$ = 6.86, p<0.001, 95% CI = 18.6% to 38.2%, Cohen's d = 2.43), as well as at the transfer location by 28.9 ± 4.9% ($t_7$ = 6.42, p<0.001, 95% CI = 18.3% to 40.0%, Cohen's d = 2.27) (*Figure 5b*). The two improvements were nearly identical ($t_7$ = 0.22, p = 0.83, 95% CI = −5.3% to 6.4%, Cohen's d = 0.08), and the TI = 1.04 ± 0.08 ($t_7$ = 11.80, p<0.001, 95% CI = 0.83 to 1.25, Cohen's d = 4.17; *Figure 5e*), indicating complete learning transfer. The learning transfer was greater than that in the training plus noise-only condition in *Figure 4d* when the transfer indices were compared (p<0.001, independent-samples Mann-Whitney U test), suggesting that top-down attention was necessary for the learning transfer.

In addition, a control condition with top-down attention trials only (without Vernier training) failed to improve the Vernier threshold at the transfer location (-1.1 ± 6.1%, $t_7$ = −0.19, p =0.85, 95% CI = −14.3% to 12.2%, Cohen's d = 0.07) (*Figure 5c*), ruling out the possibility that the learning transfer was a result of top-down attention alone. The results in *Figure 5* together indicated that, like bottom-up stimulation, top-down spatial attention to the transfer location is also sufficient to enable complete transfer of Vernier learning.

Finally, we checked the effectiveness of flashing noise suppression on Gabor perception at the transfer location. We tested twenty-three observers (eight from *Figure 4b*, five from *Figure 4e*, six from *Figure 5b*, and four from *Figure 5c*) for four blocks, 50 trials per block, after they completed their training experiments. They were asked to report whether a sub-conscious Gabor (50% of the trials) was presented at the transfer location with flashing noise suppression. The Gabor stimulus was absent in the other 50% of the trials. The mean report rate was 0.49 ± 0.01. Therefore, the results in *Figures 4,5* were not contaminated by the observers' likely perception of the Gabor stimulus at the transfer location.

## Discussion

A major focus of perceptual learning research has been on the links between learning specificity and training-induced neural changes in the brain. The common explanations include learning specificity

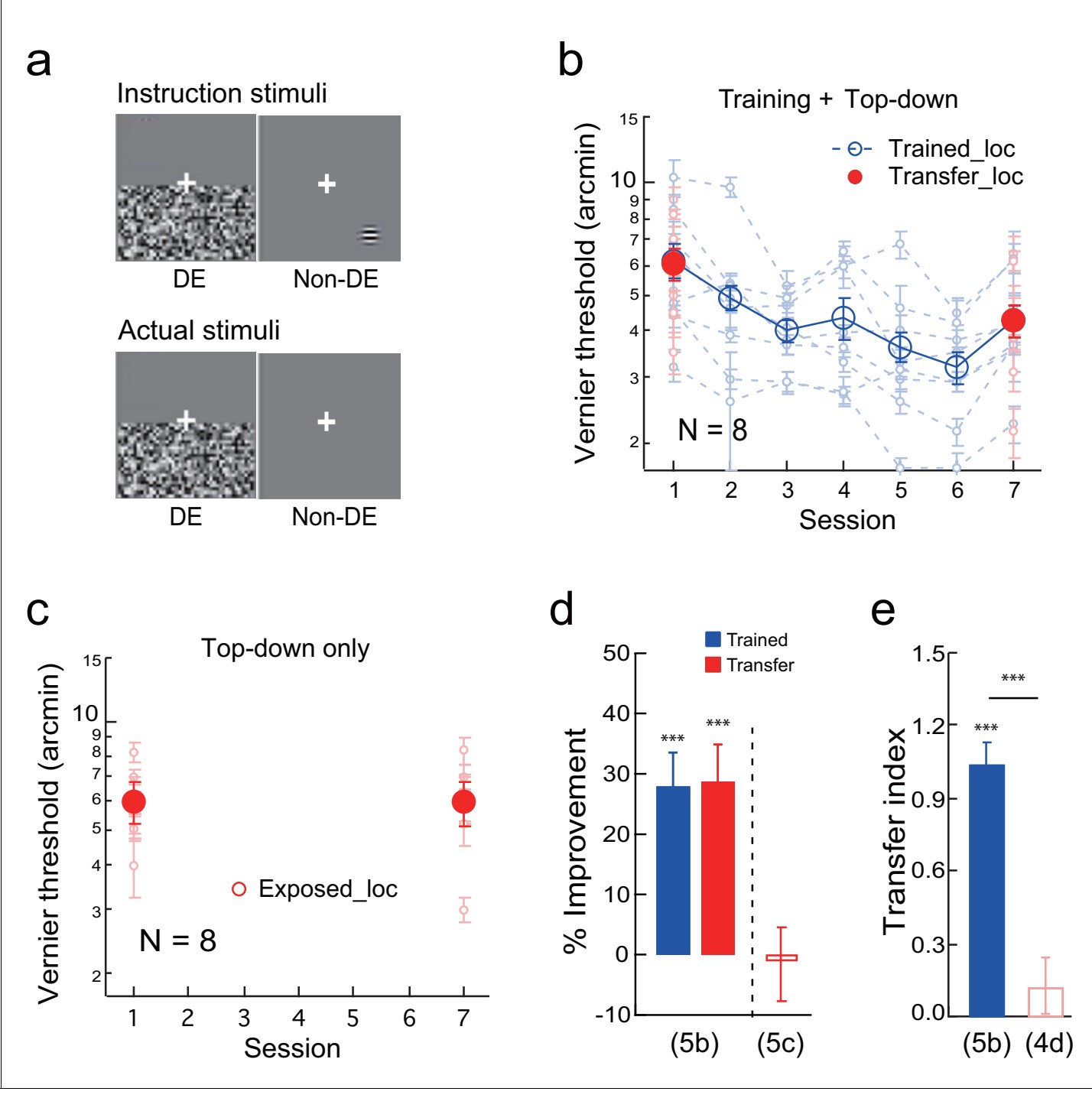

**Figure 5.** The effect of top-down spatial attention to the transfer location on Vernier learning transfer. (**a**) CFS configurations for a top-down spatial attention trial. The observers were instructed through a demo to report a Gabor or an uppercase C in the non-dominant eye at the diagonal transfer location, which was also dichoptically covered by the flashing noise in the dominant eye. In actual experiment the Gabor/C stimulus was absent. The Vernier trials and the Gabor/C trials alternated within the same block of trials. (**b**) The mean and individual learning and transfer data with training and top-down attention to the transfer location. (**c**) Control experiment. Same as 5b except that there was no Vernier training. (**d**) A summary of learning and transfer in the training plus top-down attention condition and the top-down attention alone condition. (**e**) A summary of the transfer indices in the training plus top-down attention condition and the previous training plus noise-only condition (replotted from **Figure 4d**). Error bars indicate ± 1 standard error of the mean. DE - dominant eye. *p<0.05; **p<0.01; ***p<0.001. See **Figure 5—source data 1** for raw data.

The following source data is available for figure 5:

*Figure 5 continued on next page*

*Figure 5 continued*

**Source data 1.** The first data sheet summarizes the mean and individual data presented in figure panels 5b and 5c.

as a product of neural plasticity in early visual areas that are most feature selective and retinotopic (*Karni and Sagi, 1991*; *Teich and Qian, 2003*; *Schoups et al., 2001*), or of improved readout of inputs from early visual areas (*Dosher and Lu, 1998*; *Poggio et al., 1992*; *Mollon and Danilova, 1996*; *Law and Gold, 2009*). Here our results paint a completely opposite picture: It is the actions (or the absence of actions) with the untrained conditions that decide the learning specificity and transfer. We show that for orientation and location specificity, the absence of bottom-up stimulation of neurons representing the untrained conditions, as well as top-down attention to these neurons, prevent perceptual learning from transferring to untrained conditions.

In one ERP study (*Zhang et al., 2013*), we discovered that Vernier learning and its transfer to an untrained hemisphere accompanies significant occipital P1-N1 changes when the Vernier task is performed at either the trained or the untrained location after training. However, if learning does not transfer, as shown in about half the observers, the P1-N1 changes are limited to the trained location. We interpret P1-N1 changes as possible indications of top-down connections between high-level Vernier learning and visual neurons at the trained location, as well as at untrained locations when learning transfers. We also interpret learning specificity as a result of absent functional connections. Our current data suggest that both bottom-up stimulation of, and top-down attention to, untrained conditions may activate visual neurons representing the untrained conditions, so as to foster the connections to enable learning transfer.

In another ERP study (*Zhang et al., 2015*), we also found that orientation discrimination learning and its transfer to an untrained hemisphere accompanies significant occipital C1 changes when the orientation discrimination task is performed at the trained and untrained locations, respectively, after training (*Zhang et al., 2015*). Such C1 changes are not evident with an untrained shape-discrimination task (*Zhang et al., 2015*) or with passive viewing (*Bao et al., 2010*). These results further indicate that the functional connections are task specific, consistent with the observations that perceptual learning is task specific (*Shiu and Pashler, 1992*; *Ahissar and Hochstein, 1993*; *Cong et al., 2016*). The task-specific functional connections associated with learning and its transfer are also supported by fMRI evidence, in that the generalized orientation discrimination learning is accompanied with task-specific enhancement of orientation-selective responses in the early visual areas including V1, V2 and V3 (*Byers and Serences, 2014*). These results together may also explain fMRI results that orientation specificity is accompanied with refined representation of the trained orientation in early visual areas (*Jehee et al., 2012*). This is because top-down modulation by high-level orientation learning may not reach the early cortical representations of untrained orientations as a result of absent functional connections.

As we pointed out earlier, when a near-threshold task is practiced, most brain resources are devoted to the training orientation and retinal location. The untrained orientations and retinal locations, and thus the relevant visual neurons, are neither bottom-up stimulated nor top-down attended. At least in the case of location specificity, there is evidence that spatial attention could suppress other unattended locations even with no presence of competing stimuli (*Smith et al., 2000*; *Shmuel et al., 2006*), as is typical in a perceptual learning study. We suspect that some or all of these factors could contribute to under-activations of neurons at untrained conditions, which in turn could lead to missing functional connections of these neurons to high-level learning to prevent learning transfer.

Our previous double training studies have revealed significant and often complete learning transfer to untrained conditions (*Xiao et al., 2008*; *Zhang et al., 2010*). These transfer results prompted us to propose a rule-based perceptual learning theory. Specifically, we suggest that visual perceptual learning is a high-level process. The brain learns the rules of reweighting visual inputs to achieve better visual performance, and these rules are potentially applicable to new orientations and retinal locations. More recent evidence also suggests that these rules are conceptual, in that learning can transfer between physically distinct stimuli that are initially encoded by different neural mechanisms (e.g., between local and global orientations defined by gratings and symmetric dot patterns, or between first- and second-order motion directions) (*Wang et al., 2016*). The current findings add to

this theory by elucidating why perceptual learning, if high-level and rule-based, can be specific in the first place. These results also suggest that double training schemes function by providing bottom-up and top-down forces to activate untrained neurons, which in turn initiate functional connections for learning transfer. A recent computational model explains how such functional connections can be built with double training (*Solgi et al., 2013*). In the model, the secondary training task activates the untrained neurons, which could be recalled when the brain is off-task, so that high-level 'concept neurons' that have learned the task can connect to these untrained neurons in an off-task self-organization manner for learning transfer. Apparently the activations of untrained neurons due to unconsciousness stimulation, or top-down attention without actual stimulation, can also be recalled off-task in the context of this computational account.

We once had observers practice a peripheral Vernier task identical to the current one, while flashing a Gabor simultaneously at a diagonal location (without noise suppression) for the purpose of stimulating neurons at this latter location (*Wang et al., 2012*). However, we found no evidence for learning transfer. A more recent psychophysical study also replicated the null-transfer results (*Mastropasqua et al., 2015*). Similarly, an earlier monkey-recording study (*Schoups et al., 2001*) reported that orientation discrimination learning does not transfer to a different location where a same grating stimulus is simultaneously presented. These null-transfer effects may be caused by attentional competition, in that concentrated attention to the training location would suppress the high-contrast stimuli at a different location (*Watanabe et al., 2001*). The attentional suppression effect is avoided in double training when the primary training and the secondary training that stimulates the untrained location are performed in alternating blocks of trials (*Xiao et al., 2008*), or when the stimulation of the untrained locations is below awareness as in our current study. In the latter case the bottom-up stimulation of transfer location through one eye may create an input contrast in the eye-of-origin feature, and this contrast, although invisible perceptually, has been shown to be very salient because it more strongly activates V1 neurons than the surrounding stimuli (*Zhaoping, 2008*).

Perceptual learning can also be task irrelevant. Watanabe and colleagues (*Watanabe et al., 2001*; *Seitz and Watanabe, 2003*) reported that training improves discrimination of a nearby feature that is task irrelevant and sub-threshold (to avoid attentional suppression), and that the learning occurs only when the irrelevant feature is temporally paired with rewards with the trained task. They explained this task irrelevant perceptual learning (TIPL) as a result of interactions between spatially diffusive rewards and bottom-up exposure of the task irrelevant feature (*Seitz and Watanabe, 2005*). Note that our double-training enabled learning transfer are distinct from TIPL in two important ways. First, unlike TIPL, the training and exposure do not require temporal pairing and spatial proximity of two stimuli. Indeed the training phase can precede the exposure phase with a time gap of up to 8 weeks (*Zhang et al., 2010*), and the training and exposure locations can be at diagonal quadrants of the visual field separated by 10 degrees (*Wang et al., 2012*), while double training is still effective. In the current study, the training and sub-threshold exposure are performed in separate blocks of trials, rather than paired, in *Figure 1*, and are spatially separated by 10 degrees in *Figure 4*. Second, our recent evidence suggests that double-training enabled learning transfer is still task specific (*Cong et al., 2016*). We found that orientation discrimination learning cannot transfer to a contrast discrimination task using the same Gabor stimulus, even after the observers receive additional exposure of the transfer task through easy trials in separate blocks of trials. The same is true at a reverse direction when the observers learn contrast discrimination and receive exposure of orientation discrimination, the transfer task, through easy trials.

## Materials and methods

### Observers and apparatus

One hundred and thirteen (113) undergraduate students with normal or corrected-to-normal vision participated in this study. All were inexperienced in psychophysical observations and were naïve to the purpose of the study. Informed consent, and consent to publish was obtained from each observer before testing. This study was approved by the Peking University Institution Review Board.

The stimuli were generated by a Matlab-based WinVis program (Neurometrics Institute, Oakland, CA) and presented on a 21-inch Sony G520 CRT monitor (1600 × 1200 pixel, 0.25 × 0.25 mm per

pixel, 75 Hz frame rate, 43.5 cd/m$^2$ mean luminance). The luminance of the monitor was linearized by an 8-bit look-up table. A chin-and-head rest helped stabilize the head of the observer. Experiments were run in a dimly lit room.

## Stimuli

The Gabor stimuli (Gaussian-windowed sinusoidal gratings) for foveal orientation discrimination (*Figure 1–3*) had a standard deviation at 0.48°, a spatial frequency at 1.5 cpd, a contrast at 0.47, a base orientation at 36° or 126°, and a phase randomized for every presentation. The CFS configuration consisted of a central flashing white noise pattern in the dominant eye, and sometimes a Gabor stimulus orthogonal to the trained orientation in the non-dominant eye (*Figure 1b*). The noise pattern, refreshed at 9.4 Hz, consisted of 25 × 25 randomly generated black or white blocks (0.17° × 0.17° each) for a total size of 4.30° × 4.30°. The dichoptic stimulus presentations were realized with a stereoscope. The noise pattern was presented in the dominant eye to suppress the perception of the Gabor stimulus presented in the non-dominant eye (*Tsuchiya and Koch, 2005*).

The Vernier stimuli for peripheral Vernier discrimination (*Figure 4–5*) were identical to those used in a previous study (*Wang et al., 2014*), which consisted of an upper and a lower vertical Gabor on a mean luminance screen background. The two Gabors had an identical standard deviation at 0.29°, a spatial frequency at 3 cycles per degree, a contrast of 0.47, a phase fixed at 90°, and a center to center distance at 4λ (*Figure 4a*). The vertical position of each Gabor shifted away in opposite directions to form a specific Vernier offset. The Vernier stimuli were presented in the upper left quadrant (or lower right quadrant, balanced among observers) at 5° retinal eccentricity.

The CFS configuration consisted of a flashing white noise pattern (10 Hz), which covered either the lower or upper visual hemifield opposite to the Vernier stimulus location in the dominant eye, and a horizontal Gabor in the non-dominant eye in a quadrant diagonal to the Vernier quadrant. The latter Gabor was identical to those forming the Vernier stimuli. The noise pattern consisted of 50 × 38 randomly generated black or white blocks (0.25° × 0.25° each) for a total size of 12.44° × 9.46°. The viewing distance was 1 m.

## Procedures

In an orientation discrimination trial, the fixation cross was first presented for 320 ms, and was then followed by a blank gap of 267 ms before the onset of the first stimulus interval. The reference orientation Gabor (36° or 126°, counterbalanced among observers) and the test orientation Gabor (reference + Δori) were presented in two stimulus intervals (106 ms each) in a random order, which were separated by a 533 ms inter-stimulus interval. The observers judged in which stimulus interval the Gabor was more clockwise. Auditory feedback was given on incorrect responses.

In a Vernier discrimination trial, the fixation cross was first presented for 200 ms, and was then followed by a blank gap of 200 ms before the onset of the first stimulus interval. The Vernier stimuli were then presented at one visual quadrant for 200 ms. The observers' task was to judge whether the lower Gabor was shifted to the left or the right in comparison to the upper Gabor. Auditory feedback was given on incorrect responses.

Orientation and Vernier discrimination thresholds were measured with a 2AFC staircase procedure using a classical 3-down-1-up staircase rule that resulted in a 79.4% convergence level. Each staircase consisted of four preliminary reversals and six experimental reversals (approximately 50–60 trials). The step size of the staircase was 0.05 log units. The geometric mean of the experimental reversals was taken as the threshold for each staircase run.

Each experiment consisted of seven sessions including the pre- and post-training sessions on seven different days. The pre-training session measured the orientation thresholds at the training and transfer orientations (*Figures 1–3*), or Vernier thresholds at the training and transfer locations (*Figures 4–5*), for six staircases each. The post-training session measured the same thresholds for five staircases each. The geometric mean was taken as the pre- or post-training threshold with each condition. The five training sessions each consisted of 10 staircases of orientation discrimination training (*Figures 1–3*) or Vernier discrimination (*Figures 4–5*), as well as 10 blocks of bottom-up and/or top-down trials with the transfer condition (50 trials per block in *Figures 1–3*, or the same number of trials as in 10 training staircases in *Figures 4–5*), with the training and bottom-up/top-down tasks switched every five blocks of trials. Each training session lasted about 1.5 hr.

## Acknowledgements

This research was supported by a Natural Science Foundation of China Grant 31230030 (CY). We thank the helpful comments and suggestions from Fang Fang, Sheng He, Stanley Klein, Wu Li, Shin Shimojo, Rufin Vogels, and Li Zhaoping during various stages of this study.

## Additional information

### Funding

| Funder | Grant reference number | Author |
| --- | --- | --- |
| National Natural Science Foundation of China | 31230030 | Cong Yu |

The funders had no role in study design, data collection and interpretation, or the decision to submit the work for publication.

### Author contributions

Y-ZX, Conception and design, Acquisition of data, Analysis and interpretation of data, Drafting or revising the article; J-YZ, CY, Conception and design, Analysis and interpretation of data, Drafting or revising the article

### Author ORCIDs

Cong Yu, http://orcid.org/0000-0002-8453-6974

### Ethics

Human subjects: Informed consent, and consent to publish was obtained from each observer before testing. This study was approved by the Peking University Institution Review Board.

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
