## [Decision Letter]

[Editors’ note: this article was originally rejected after discussions between the reviewers, but the authors were invited to resubmit after an appeal against the decision.]

Thank you for submitting your work entitled "Bottom-up and top-down influences at untrained conditions determine perceptual learning specificity and transfer" for consideration by *eLife*. Your article has been reviewed by two peer reviewers, and the evaluation has been overseen by a Reviewing Editor and David Van Essen as the Senior Editor. Our decision has been reached after consultation between the reviewers. Based on these discussions and the individual reviews below, we regret to inform you that this submission will not be considered further for publication in *eLife*.

Although the reviewers and the Reviewing Editor were in general agreement that the topic of the study is quite interesting and could potentially advance our understanding of specificity and transfer in perceptual learning, they also felt that the study lacked critical controls that are needed to validate the main findings. Specifically, it would be important to show if and how much learning occurs in the bottom-up and top-down conditions for subjects that do not also go through the training sequence. If you are able to carry out these controls and address the concerns, we invite you to resubmit this work to *eLife*.

The reviewers also noted the following other major concerns that you may want to keep in mind for possible revisions:

1) Given that the authors' previous work has already shown that some activity is needed to induce transfer, it is important to more clearly describe the novelty of the current results to a general-audience journal.

2) Across the experiments, the number of subjects is quite variable, with E1 having double the subjects as some other studies (N=10 vs N=8 vs N=5, etc.). This is an important issue, especially since transfer effects are compared across subject groups and the number of subjects for individual groups is relatively low to begin with. What criteria were used to determine how many subjects were run in each study and when data collection in each study should be stopped?

3) There are a number of issues with the statistical methods that should be addressed, including: a) specifying the meaning of error bars in the figures and reported measurement/sampling variability in the text (SEM? STD? Confidence intervals?); b) the transfer index is used in the text but not in the figures to summarize the findings, and a significance test of the value of the transfer index is reported only for the data presented in Figure 1 and not the other Figures, and no transfer index is reported for the spatial experiments; c) many non-pairwise t-tests were performed (non-parametric tests would probably be more appropriate) without correction for multiple comparisons – most were highly significant, but this procedure should still be implemented; and d) Do any of the statistical tests take into account the uncertainty in the estimates of the various values measured per subject (e.g., the error bars apparent in the dim points in Figure 1, etc.)?

4) In the CSF experiments, the authors mention that the subjects were 'neither told nor aware of' the untrained stimuli in the non-dominant eye (NDE). Were any queries made of the subjects to confirm that the stimulus presented to the NDE was actually 'subconsciously' perceived?

*Reviewer #1:*

In this study, the authors investigated effects of bottom-up and top-down stimulation of the untrained condition in determining specificity and transfer in perceptual learning. The continuous flash suppression (CFS) technique was used to present subconscious bottom-up or top-down stimuli to the transfer orientation or location. They found that either bottom-up stimulation or top-down attention is sufficient to enable significant learning transfer, and conclude that learning specificity may result from under activation of untrained visual neurons due to insufficient bottom-up stimulation and/or top-down attention.

The topic of the study is quite interesting and could potentially advance our understanding of specificity and transfer in perceptual learning. However, it seems to me that the study may have missed a critical control condition – how much learning was induced by the bottom-up and top-down stimulation of the transfer orientation or location? Although the results are presented as transfer of learning from the trained condition, stimulation of the transfer condition could by itself lead to improved performance.

*Reviewer #2:*

In this paper, Xiong et al. show that both bottom-up and top-down factors can amplify transfer effects to untrained stimulus orientations and locations. Their results provide one possible explanation regarding why transfer effects have been inconsistently reported in the past. I think the experiments are generally well designed and well controlled. However, I have several concerns that I would like to see addressed. Moreover, the authors’ previous work has already shown that some activity is needed to induce transfer. So the novelty of the present demonstration that either bottom-up or top-down activation is sufficient should be clarified for a general audience journal.

Specific comments

1) Number of subjects: Across the experiments, the number of subjects is quite variable, with E1 having double the subjects as some other studies (N=10 vs N=8 vs N=5, etc.). This is an important issue, especially since transfer effects are compared across subject groups and the number of subjects for individual groups is relatively low to begin with. This left me wondering what criteria were used to determine how many subjects were going to be run in each study and when data collection in each study should be stopped.

2) Statistical methods. The authors used a transfer index for the baseline experiment to compare the transfer effects between experimental conditions (which is fine). The authors then provide statistical results for Figure 1. But for other comparisons, statistical results comparing transfer indexes between the baseline (Figure 1) experimental conditions in Figure 2–Figure 3 were not reported. Also many non-pairwise t-tests were performed without correction for multiple comparisons – true most were highly significant, but this procedure should still be implemented. For the spatial experiments, no formal transfer indexes were established.

3) In the CSF experiments, the authors mention that the subjects were 'neither told nor aware of' the untrained stimuli in the non-dominant eye (NDE). Were any queries made of the subjects to confirm that the stimulus presented to the NDE was actually 'subconsciously' perceived?

[Editors’ note: what now follows is the decision letter after the authors submitted for further consideration.]

Thank you for resubmitting your work entitled "Bottom-up and top-down influences at untrained conditions determine perceptual learning specificity and transfer" for further consideration at *eLife*. Your revised article has been favorably evaluated by David Van Essen (Senior editor), a Reviewing editor, and two reviewers.

The manuscript has been much improved, but there are some remaining minor issues that should be addressed before acceptance, as outlined below:

1) Figure 4 appear to be mislabeled. The last panel should be (d) not (e) – so in the summary figures I think it would be better to indicate in all bar graphs which figures they derive from. For example, in Figure 4 – the label should be (4b) (4c) and (4d) instead of (b) (c) and (e). Another example is Figure 3 – the label could be (3b) (1a) and (1f) instead of (b) (1a) (1f).

2) The authors should consider discussing the Jehee J. Neuro and Byers J. Neurophys papers given the relevance of top down attention during training. Another potentially relevant paper is Szpiro and Carrasco (2015) in Psych Sci, which is seemingly both consistent and inconsistent with the present findings – they showed that exogenous attention induces transfer in spatial frequency but not an orientation task.

3) All raw source data should be uploaded. Currently it appears that only a summary file is provided in. xls format.

---

## [Author Response]

[Editors’ note: the author responses to the first round of peer review follow.]

*Although the reviewers and the Reviewing Editor were in general agreement that the topic of the study is quite interesting and could potentially advance our understanding of specificity and transfer in perceptual learning, they also felt that the study lacked critical controls that are needed to validate the main findings. Specifically, it would be important to show if and how much learning occurs in the bottom-up and top-down conditions for subjects that do not also go through the training sequence. If you are able to carry out these controls and address the concerns, we invite you to resubmit this work to eLife.*

These control data are now presented in Figure 1,Figure 2,Figure 3,Figure 4,Figure 5. Among these control conditions, Figure 1,Figure 2,Figure 4,Figure 5 show no significant effects of the bottom-up or top-down conditions on learning at the transfer condition when no training is carried out. Figure 3 shows a small learning effect that is significantly lower than the improvement with the training sequence (14.6% vs. 45.1%, p = 0.001). These control results thus validate the main findings in this study.

*The reviewers also noted the following other major concerns that you may want to keep in mind for possible revisions:*

*1) Given that the authors' previous work has already shown that some activity is needed to induce transfer, it is important to more clearly describe the novelty of the current results to a general-audience journal.*

We describe the novelty and significant of this study to the general audience more clearly at the end of the Introduction:

“These results provide a solution to the mystery of learning specificity that has dominated the history of perceptual learning research. With learning specificity considered as a by-product of training, the field should move on to study the brain mechanisms of perceptual learning without much of specificity-related constraints. Moreover, more efficient training paradigms can be designed to generate perceptual learning without the unwanted specificity in practical settings.”

2) Across the experiments, the number of subjects is quite variable, with E1 having double the subjects as some other studies (N=10 vs N=8 vs N=5, etc.). This is an important issue, especially since transfer effects are compared across subject groups and the number of subjects for individual groups is relatively low to begin with. What criteria were used to determine how many subjects were run in each study and when data collection in each study should be stopped?

We did a power analysis in a recent JOV paper (Cong et al., 2016) and found that if only the pre- and post-training thresholds are compared with a t-test, 5-6 subjects is sufficient to obtain high statistical power in a perceptual learning experiment. However, this number is inadequate when multiple comparisons are performed. We thus increased the number of subjects in each experiment to 8-10 to deal with the power issue.

*3) There are a number of issues with the statistical methods that should be addressed, including: a) specifying the meaning of error bars in the figures and reported measurement/sampling variability in the text (SEM? STD? Confidence intervals?);*

SEM is specified in the legend of each figure; SEM and 95% confidence levels are specified in the text.

*b) The transfer index is used in the text but not in the figures to summarize the findings, and a significance test of the value of the transfer index is reported only for the data presented in Figure 1 and not the other Figures, and no transfer index is reported for the spatial experiments;*

The transfer indices are now summarized in each figure. A significance test is also reported for each transfer index.

*c) Many non-pairwise t-tests were performed (non-parametric tests would probably be more appropriate) without correction for multiple comparisons – most were highly significant, but this procedure should still be implemented;*

Independent-samples Kruskal-Wallis test with correction for multiple comparisons (Figure 1–Figure 3) and Mann-Whitney U test (Figure 4–Figure 5) are now performed to compare transfer indices among different conditions.

*d) Do any of the statistical tests take into account the uncertainty in the estimates of the various values measured per subject (e.g., the error bars apparent in the dim points in Figure 1, etc.)?*

Our statistical tests do not consider the error bars of individual data. We consulted an expert in statistics on this issue, and the response is: if the mean is calculated when each individual datum is weighted by its standard error, there is a risk that when one datum has very small error bars, the calculated mean will be strongly biased by this datum.

*4) In the CSF experiments, the authors mention that the subjects were 'neither told nor aware of' the untrained stimuli in the non-dominant eye (NDE). Were any queries made of the subjects to confirm that the stimulus presented to the NDE was actually 'subconsciously' perceived?*

Yes, we not only asked the observers after the experiments, but also tested some of them to confirm their oral reports. These tests are described in two occasions of the text:

“In Figure 3b-c the Gabor/C judgments were near chance (0.80), suggesting effective suppression of the perception of the orthogonal Gabor by the continuous flashing noise. We further confirmed this CFS effect with 18 observers (10 from Figure 1, 3 from Figure 2, and 5 from Figure 3) after they completed their experiments. Each observer was presented a sub-conscious Gabor at the transfer orientation in half the trials with flashing noise presented to the other eye (60 trials per block for 4 blocks). In the other half trials a blank screen was presented. The result showed a chance-level accuracy at 0.51 ± 0.01. Therefore, the transfer results in Figure 1–Figure 3 were not contaminated by the leakage of the Gabor perception.”

“Finally, we checked the effectiveness of flashing noise suppression on Gabor perception at the transfer location. We tested observers (13 from Figure 4, 4 from Figure 4, and 6 from Figure 5) for 4 blocks, 50 trials per block, after they completed their training experiments. They were asked to report whether a sub-conscious Gabor (50% of the trials) was presented at the transfer location with flashing noise suppression. The Gabor stimulus was absent in the other 50% of the trials. The mean report rate was 0.49 ± 0.01. Therefore, the results in Figure 5 were not contaminated by the observers’ likely perception of the Gabor stimulus at the transfer location.”

*Reviewer #1:*

*In this study, the authors investigated effects of bottom-up and top-down stimulation of the untrained condition in determining specificity and transfer in perceptual learning. The continuous flash suppression (CFS) technique was used to present subconscious bottom-up or top-down stimuli to the transfer orientation or location. They found that either bottom-up stimulation or top-down attention is sufficient to enable significant learning transfer, and conclude that learning specificity may result from under activation of untrained visual neurons due to insufficient bottom-up stimulation and/or top-down attention.*

The topic of the study is quite interesting and could potentially advance our understanding of specificity and transfer in perceptual learning. However, it seems to me that the study may have missed a critical control condition – how much learning was induced by the bottom-up and top-down stimulation of the transfer orientation or location? Although the results are presented as transfer of learning from the trained condition, stimulation of the transfer condition could by itself lead to improved performance.

These control data are now presented in Figure 1,Figure 2,Figure 3,Figure 4,Figure 5. Among these control conditions, Figure 1,Figure 2,Figure 4,Figure 5 show no significant effects of the bottom-up or top-down conditions on learning at the transfer condition when no training is carried out. Figure 3 shows a small learning effect that is significantly lower than the improvement with the training sequence (14.6% vs. 45.1%, p = 0.001). These control results thus validate the main findings in this study.

*Reviewer #2:*

*In this paper, Xiong et al. show that both bottom-up and top-down factors can amplify transfer effects to untrained stimulus orientations and locations. Their results provide one possible explanation regarding why transfer effects have been inconsistently reported in the past. I think the experiments are generally well designed and well controlled. However, I have several concerns that I would like to see addressed. Moreover, the authors’ previous work has already shown that some activity is needed to induce transfer. So the novelty of the present demonstration that either bottom-up or top-down activation is sufficient should be clarified for a general audience journal.*

This issue is addressed in the revision. See our responses above.

*Specific comments*

1) Number of subjects: Across the experiments, the number of subjects is quite variable, with E1 having double the subjects as some other studies (N=10 vs N=8 vs N=5, etc.). This is an important issue, especially since transfer effects are compared across subject groups and the number of subjects for individual groups is relatively low to begin with. This left me wondering what criteria were used to determine how many subjects were going to be run in each study and when data collection in each study should be stopped.

This issue is addressed in the revision. See our responses above.

*2) Statistical methods. The authors used a transfer index for the baseline experiment to compare the transfer effects between experimental conditions (which is fine). The authors then provide statistical results for Figure 1. But for other comparisons, statistical results comparing transfer indexes between the baseline (Figure 1) experimental conditions in Figure 2–Figure 3 were not reported. Also many non-pairwise t-tests were performed without correction for multiple comparisons – true most were highly significant, but this procedure should still be implemented. For the spatial experiments, no formal transfer indexes were established.*

These issues are addressed in the revision. See our responses above.

*3) In the CSF experiments, the authors mention that the subjects were 'neither told nor aware of' the untrained stimuli in the non-dominant eye (NDE). Were any queries made of the subjects to confirm that the stimulus presented to the NDE was actually 'subconsciously' perceived?*

This issue is addressed in the revision. See our responses above.

[Editors’ note: the author responses to the re-review follow.]

*The manuscript has been much improved, but there are some remaining minor issues that should be addressed before acceptance, as outlined below:*

*1) Figure 4 appear to be mislabeled. The last panel should be (d) not (e) – so in the summary figures I think it would be better to indicate in all bar graphs which figures they derive from. For example, in Figure 4 – the label should be (4b) (4c) and (4d) instead of (b) (c) and (e). Another example is Figure 3 – the label could be (3b) (1a) and (1f) instead of (b) (1a) (1f).*

Thanks a lot for pointing out the mistakes. The labels have been updated as suggested.

2) The authors should consider discussing the Jehee J. Neuro and Byers J. Neurophys papers given the relevance of top down attention during training. Another potentially relevant paper is Szpiro and Carrasco (2015) in Psych Sci, which is seemingly both consistent and inconsistent with the present findings – they showed that exogenous attention induces transfer in spatial frequency but not an orientation task.

We added the following sentences to an existing paragraph in the Discussion section to discuss the Jehee paper (Jehhe et al., 2012) and the Byers paper (Byers and Serences, 2014):

“In another ERP study (Zhang et al., 2015), we also found that orientation discrimination learning and its transfer to an untrained hemisphere accompanies significant occipital C1 changes when the orientation discrimination task is performed at the trained and untrained locations, respectively, after training (Zhang et al., 2015). Such C1 changes are not evident with an untrained shape-discrimination task (Zhang et al., 2015) or with passive viewing (Bao et al., 2010). These results further indicate that the functional connections are task specific, consistent with the observations that perceptual learning is task specific (Shiu and Pashler, 1992; Ahissar and Hochstein, 1993; Cong et al., 2016). The task-specific functional connections associated with learning and its transfer are also supported by fMRI evidence, in that the generalized orientation discrimination learning is accompanied with task-specific enhancement of orientation-selective responses in the early visual areas including V1, V2 and V3 (Byers and Serences, 2014). These results together may also explain fMRI results that orientation specificity is accompanied with refined representation of the trained orientation in early visual areas (Jehee et al., 2012). This is because top-down modulation by high-level orientation learning may not reach the early cortical representations of untrained orientations as a result of absent functional connections.”

Szpiro and Carrasco (2015, Psych Sci) reported that three days of training (800 trials per day) failed to improve the performance of an orientation comparison task, unless exogenous attention was added. However, 2400 trials in 3 days is a lot of trials for perceptual learning to occur in many tasks. So we double-checked their non-learning baseline results with nearly identical procedures. All observers we trained showed consistent and significant orientation learning, as much as when exogenous attention was introduced. We showed the data to Carrasco recently when she was visiting us in Beijing. It would be better that we solve the discrepancies with Carrasco in a different occasion.

*3) All raw source data should be uploaded. Currently it appears that only a summary file is provided in. xls format.*

Now we upload five Excel files, each containing data for one figure. Each data sheet in an Excel file contains data for one specific subpanel of the figure. Each observer’s raw data (individual staircase data with values of reversals), as well as the calculations for the individual means and group means are listed.